# Detection of *Schistosoma mekongi* DNA in Human Stool and Intermediate Host Snail *Neotricula aperta* via Loop-Mediated Isothermal Amplification Assay in Lao PDR

**DOI:** 10.3390/pathogens11121413

**Published:** 2022-11-24

**Authors:** Takashi Kumagai, Emilie Louise Akiko Matsumoto-Takahashi, Hirofumi Ishikawa, Sengdeuane Keomalaphet, Phonepadith Khattignavong, Pheovaly Soundala, Bouasy Hongvanthong, Kei Oyoshi, Yoshinobu Sasaki, Yousei Mizukami, Shigeyuki Kano, Paul T. Brey, Moritoshi Iwagami

**Affiliations:** 1Department of Parasitology and Tropical Medicine, Tokyo Medical and Dental University, Tokyo 113-8510, Japan; 2Research Institute, National Center for Global Health and Medicine, Tokyo 162-8655, Japan; 3Graduate School of Public Health, St. Luke’s International University, Tokyo 104-0044, Japan; 4Institut Pasteur du Laos, Ministry of Health, Vientiane P.O. Box 3560, Laos; 5Center of Malariology, Parasitology and Entomology, Ministry of Health, Vientiane P.O. Box 0100, Laos; 6Earth Observation Research Center, Japan Aerospace Exploration Agency (JAEA), Tsukuba 305-8505, Japan

**Keywords:** *Schistosoma mekongi*, LAMP, Laos, risk map

## Abstract

Schistosomiasis mekongi infection represents a public health concern in Laos and Cambodia. While both countries have made significant progress in disease control over the past few decades, eradication has not yet been achieved. Recently, several studies reported the application of loop-mediated isothermal amplification (LAMP) for detecting *Schistosoma* DNA in low-transmission settings. The objective of this study was to develop a LAMP assay for *Schistosoma mekongi* using a simple DNA extraction method. In particular, we evaluated the utility of the LAMP assay for detecting *S. mekongi* DNA in human stool and snail samples in endemic areas in Laos. We then used the LAMP assay results to develop a risk map for monitoring schistosomiasis mekongi and preventing epidemics. A total of 272 stool samples were collected from villagers on Khon Island in the southern part of Laos in 2016. DNA for LAMP assays was extracted via the hot-alkaline method. Following the Kato-Katz method, we determined that 0.4% (1/272) of the stool samples were positive for *S. mekongi* eggs, as opposed to 2.9% (8/272) for *S. mekongi* DNA based on the LAMP assays. Snail samples (*n* = 11,762) were annually collected along the riverside of Khon Island from 2016 to 2018. DNA was extracted from pooled snails as per the hot-alkaline method. The LAMP assay indicated that the prevalence of *S. mekongi* in snails was 0.26% in 2016, 0.08% in 2017, and less than 0.03% in 2018. Based on the LAMP assay results, a risk map for schistosomiasis with kernel density estimation was created, and the distribution of positive individuals and snails was consistent. In a subsequent survey of residents, schistosomiasis prevalence among villagers with latrines at home was lower than that among villagers without latrines. This is the first study to develop and evaluate a LAMP assay for *S. mekongi* detection in stools and snails. Our findings indicate that the LAMP assay is an effective method for monitoring pathogen prevalence and creating risk maps for schistosomiasis.

## 1. Introduction

Schistosomiasis is a neglected tropical disease (NTD), occurring mainly in tropical and sub-tropical regions, which is associated with the conditions of poverty in developing countries [1]. *Schistosoma mekongi*, the causative pathogen, is distributed in the southern part of the Lao People’s Democratic Republic (Laos) and the northern part of Cambodia, along the Mekong River. A risk of infection with *S. mekongi* was reported for around 50,000 households, comprising an estimated 150,000 people in Laos and Cambodia [2]. The intermediate host snail, *Neotricula aperta*, is frequently found in the Mekong River [3]. In particular, a more widespread larger-river dwelling form is found in southern Laos and northern Cambodia [4]. There are three strains of *N. aperta* (α-, β- and γ-strains), among which the γ-strain is the only strain capable of transmitting *S. mekongi* in nature [3]. The period of highest transmission, March–April, coincides with the dry season when water levels are low, with the observed snail populations reaching their maximum [3]. Most residents of the region depend on fishing and use the river water for personal hygiene and occupational, and recreational activities [3]. Schistosome re-infection has severe consequences and is associated with various complications in endemic areas [5].

In Laos, schistosomiasis mekongi has been endemic in the Khong (152 villages) and Mounlapamok (50 villages) districts of the Champasak Province. To control the disease, six rounds of preventative chemotherapy (or mass drug administration: MDA) with praziquantel were implemented in the endemic areas of Laos during the period of 1989–1998 [6]. As a result, the infection rate drastically decreased to an average of 2.1% in the Khong and 0.4% in the Mounlapamok District [6]. However, a field survey conducted by the Lao Ministry of Health with support from the WHO in 2003 revealed that the infection rate had increased again, to an average of 11% (0–47.2%) across 64 endemic communities in the Khong District [7]. The average infection rate in the Mounlapamok District was 0.7%. As a result, the Lao MoH resumed MDA to the high-risk population (5 to 60 years of age) in the two endemic districts in 2007 [3,7].

In 2017, the Lao MoH and WHO adopted goals and indicators for the elimination of schistosomiasis from the Western Pacific Region [8]. These are as follows: (a) the reduction in human infections to zero, (b) the reduction in animal infections to zero, and (c) the reduction in snail infections to zero, all by 2025. These goals are expected to eliminate schistosomiasis by 2030. In 2017, infection prevalence in Laos was below 3%, with only 0.1% being high-intensity infections. In 2018, the infection prevalence was 3.2%, but there were no patients with high-intensity infection [7].

The monitoring systems for schistosomiasis mekongi currently employed in Laos are insufficient. In fact, up-to-date information on the areas at risk of schistosomiasis is lacking. An effective monitoring system is of utmost necessity for endemic areas. Effective surveillance systems should be implemented after eliminating schistosomiasis so as to monitor and prevent its re-introduction into endemic areas [9]. The Kato–Katz microscopy-based method has been used as the golden standard for detecting *S. mekongi* eggs in human stool samples as a means of disease monitoring. However, the sensitivity of this method is not sufficient for detecting minor infections, which are currently predominant among patients in the endemic areas in Laos. A highly sensitive diagnostic method is, thus, urgently needed in order to monitor the prevalence and evaluate interventions in the region.

Molecular diagnostic methods for schistosomiasis mekongi have been reported using real-time PCR, but infection experiments have been performed in a laboratory and have not been tested in the field [10]. In addition, DNA detection from serum has also been reported, but the sensitivity is not sufficient [11]. Several studies have reported that a novel molecular method, the loop-mediated isothermal amplification (LAMP) assay, is useful for the detection and monitoring of schistosomiasis in endemic areas [12,13,14,15,16,17,18]. The LAMP method enabled detection in low-transmission areas and can be applied for point-of-care testing in resource-limited settings [19,20,21].

In the present study, we developed a LAMP assay for detecting the ITS1 region of *S. mekongi* ribosomal DNA and evaluated its detection capacity in human stool samples, comparing it to the Kato–Katz method. We also examined infection prevalence in the intermediate host snail *N. aperta*, collected from riversides in the endemic areas where the human stools were also collected. Based on the distribution of schistosomiasis patients and infected snails, we created a heat map that shows the infection risk within the region. 

## 2. Materials and Methods

### 2.1. Study Area

One human field survey in April 2016 and three snail field surveys in April 2016, 2017, and 2018 were conducted in Don Khon, that is, Khon Island, Khong District and Champasak Province, Laos (Figure 1). In 2016, the total of villagers of each of the villages in Don Khon, Khon Neua and Khon tai in the north was 1441 (809 males and 632 females) in 234 households, while the total of villagers of Hang Khon in the south was 366 (246 males and 116 females) in 55 households. These two areas cover most of the island’s population. This island was chosen as the study area due to a high prevalence of schistosomiasis in a preliminary survey in 2015, despite the annual MDA with praziquantel. Surveys were carried out at the end of the dry season because this is the only time when snail collection is possible. 

### 2.2. Human Stool Sampling

A total of 272 villagers from the 3 villages voluntarily participated in the study, which was announced to them by village chiefs (Figure 1). Our study team was based at two temples (Khon Neua and Khon Tai) and a school (Hang Khon) where the participants would visit us. An interview survey about lifestyle habits related to parasite infection was conducted by our team using a questionnaire form. A stool container was given to each participant after the survey, and a single stool sample was collected from each participant on the next day. Each stool sample was fixed with 70% EtOH on site, after making three slides of Kato–Katz thick smears for microscopy analysis. Samples for LAMP assays were stored at room temperature. 

### 2.3. Snail Sampling

Three snail field surveys were conducted on Khon Island in April 2016, 2017, and 2018. The intermediate host snail *N. aperta* was collected at several sites along the riverside of the island (Figure 1). *N. aperta* was morphologically identified using stereoscopic microscopy. The total number of collected *N. aperta* individuals was 2862, 5900, and 3000 in 2016, 2017, and 2018, respectively. All host snail samples were fixed with 70% EtOH on site and stored at room temperature for subsequent LAMP assays. 

All stool and snail samples were transported to Tokyo Medical and Dental University, Japan, after obtaining authorization for shipment from the Lao MoH.

### 2.4. Parasitological Tests Using the Kato–Katz Method

Three slides of the Kato–Katz thick smear were prepared from each stool sample using a commercially available Kato–Katz Stool Examination Kit (Mahidol University, Nakhon Pathom, Thailand), according to the manufacturer’s instructions. The slides were observed by experienced technicians under a microscope in order to detect *S. mekongi* eggs. Approximately 40 mg of stool was mounted on each slide. When one of the three slides was positive for *S. mekongi* eggs, the participant was considered schistosomiasis-positive, that is, a schistosomiasis mekongi patient. 

### 2.5. DNA Extraction

For stool samples, an easy heat alkaline DNA extraction method was applied as described in a previous study, with slight modifications [12,22]. Briefly, 500 mg of the EtOH-fixed stool sample was collected from a stool container and washed with distilled water. The sample was heated in 1 mL of 50 mM NaOH at 95 °C for 1 h by vortexing the sample several times. After centrifugation at 15,000 rpm for 5 min, the supernatant was collected and diluted with distilled water 20 times the volume. The diluted DNA solution was immediately used as the template for LAMP assays.

The same DNA extraction method was used for host snail samples. In brief, the EtOH-fixed snails were washed with distilled water in a 50 mL tube. Pools of snails (50 or 200 snails per group) were crushed with scissors in 5 mL of 50 mM NaOH. The snail solution was heated to 95 °C for 1 h and centrifuged at 15,000 rpm for 5 min. The supernatant (DNA solution) was collected in 1.5 mL tubes and stored at −30 °C, until the LAMP assay was performed.

### 2.6. LAMP Assay

The ITS-1 region of the *S. mekongi* ribosomal DNA gene (GenBank accession no. U82284.1) was selected as the target sequence for LAMP detection. We designed specific primer sets (Appendix A) using PrimerExplorer V5 software on the website of Eiken Chemical, Inc., Tokyo, Japan. The LAMP assay was performed using Loopamp™ DNA Amplification Reagent and Fluorescent Detection Reagent, according to the manufacturer’s instructions (Eiken Science, Tokyo, Japan). The reaction tubes were incubated at a constant temperature of 63 °C for 60 min and then at 85 °C for 5 min to stop the reaction. Positive (*S. mekongi* DNA) and negative (water) controls were used in each assay. Amplification of the target gene was confirmed using a fluorescence dye under UV.

### 2.7. Heat Mapping

High-resolution satellite images (110 km^2^) of Khon Island, Khong District and Champasak Province, Laos, were purchased from the JAPAN SPACE IMAGING CORPORATION (Tokyo, Japan; https://www.jsicorp.jp/en.html, accessed on 10 October 2022). The houses of the patients diagnosed via LAMP assays were plotted using a Quantum Geographic Information (QGIS) System 2.18.22 (Development Team-Open-Source Geospatial Foundation Project, 2018). We also plotted the locations at which infected snails were collected. Using the QGIS heat-map plugin, we created heat maps of *S. mekongi*-positive snails and humans using LAMP assays. The plugin is based on kernel density, which smoothly estimates the whole distribution of specimens. JPMAP utilizes GSMaP [23] provided by JAXA as the precipitation data and the MOD11/MYD11 products [24] provided by the National Aeronautics and Space Administration (NASA)/United States Geological Survey (USGS) as the land surface temperature (LST) data. JPMAP allows users to search for and obtain data within a selected period or area (point, a rectangle administrative area) via a user-friendly website (https://www.jpmap-jaxa.jp/jpmap/en/, accessed on 10 October 2022).

### 2.8. Statistical Analysis

Assessment of the *S. mekongi* infection rate in the host snails was followed by an exact binomial test. Fisher’s exact test and ANOVA were used to identify an association between the lifestyle of participants, including their socioeconomic status, and infection status, based on LAMP assay results. All statistical analyses were conducted using SPSS version 24.0 (SPSS Inc., Chicago, IL, USA).

## 3. Results

### 3.1. Sensitivity and Specificity of LAMP Assay Targeting S. mekongi

To date, only direct microscopic detection methods have been used to confirm *S. mekongi* infections in humans and snails. In this study, we developed a new approach for detecting *S. mekongi* infection using the LAMP method and created an infection risk map based on assay results. To achieve high sensitivity and specificity, we developed primers that target the ITS1 region of *S. mekongi*. The LAMP assay detection limit was 1 pg of *S. mekongi* DNA per reaction (Figure 2). We also confirmed that the LAMP assay did not amplify *S. mansoni* nor *S. japonicum* DNA (Figure 2). Such sensitivity is sufficient for the detection of a single miracidium of schistosomes. The specificity was also high, as the ITS1 DNA sequences of schistosomes are quite different from those of other flukes (e.g., *Opisthorchis viverrini* and *Haplorchis taichui*), which are prevalent in the same endemic areas of the Khong District, Laos. 

### 3.2. Comparison of LAMP Detection and the Kato–Katz Method in Stool Samples

To investigate the utility of LAMP detection in the field, fecal samples were collected from residents of Don Khon, an endemic area for schistosomiasis mekongi in Lao PDR. In 2016, feces were collected from approximately 100 residents in 3 villages within Don Khon and were then subjected to detection via the Kato–Katz method (Figure 1). DNA was then extracted from the remaining feces and detected using the LAMP method. We employed an easy and cost-effective DNA extraction method, namely heat–NaOH DNA extraction [12,14,22], since we designed the LAMP system to be utilized for point-of-care testing in resource-limited settings. 

Only 0.4% (1/272) of the stool samples from the Khon Tai village tested positive for *S. mekongi* eggs using the Kato–Katz method (Table 1). In contrast, 55.9% (152/272) of the stool samples were positive for eggs of the liver fluke *O. viverrini,* and 22.8% (62/210) were positive for hookworms (the species was not identified) (Table 1). 

The LAMP assay results are shown in Table 1. In the Khon Neua village, 4.2% (4/96) of the stool samples were positive for *S. mekongi* DNA detected via the LAMP assay, whereas all the samples were negative based on Kato–Katz detection. In the Khon Tai village, 3.4% (3/89) of stool samples were positive for *S. mekongi* DNA versus 1.1% (1/89) via the Kato–Katz method. The Kato–Katz positive samples were also positive for *S. mekongi* DNA. In Hang Khon, 1.1% (1/87) of stool samples were positive for *S. mekongi* DNA, while the Kato–Katz method did not detect any eggs. Overall, 2.9% (8/272) of the stool samples were positive for *S. mekongi* DNA, as determined via the LAMP assay. Therefore, the LAMP assay exhibited sensitivity (2.9%) that was much higher than that of the Kato–Katz method (0.4%).

### 3.3. Large-Scale Monitoring of Snail Samples via LAMP Detection

Next, to determine the infection rates among intermediate host snails near the residential area, 2862 snails were collected at 6 breeding sites on the northeastern riverside in Don Khon during April 2016 (Table 2 and Figure 1). DNA was extracted from pooled host snails (86, 176, and 200 snails per group). LAMP assays were employed for the detection of *S. mekongi* DNA in the snails collected from three breeding sites (P2, P4, and P5) (Table 2). At the P2 breeding site, three positive groups were found among the seven groups. The infection rate of snails at the P2 site was calculated as 0.28%, with a 95% CI of 0.07–0.74%. Precise assessment of the infection rate could be achieved via one-by-one inspections of all snails, which would be impractical in the case of a large snail population with extremely low infection rates, as in the current study. Such cases require an alternative method for efficient infection rate estimation. This method examines a pool of snails that comprise a certain number (g) of snails crushed together, where the group composition number g satisfies the following condition: 1-(1-*p* max) g < 0.5 (*p* max being the assessed upper limit of the infection). In this area, the upper limit of *S. mekongi* infection in the snails (*p* max) was assessed to be 1% beyond the upper limit of 95% CI at the P2 breeding site in 2017, so that g = 50, that is, 50 snails were crushed together as a group, which satisfies the condition, as previously reported in the case of *S. japonicum* [13]. In 2017, at the P2 and P4 breeding sites, 1 positive group was found among 20 groups (1000 snails in total) and 12 groups (600 snails), respectively. Therefore, the infection rates of the P2 and the P4 sites were estimated at 0.10% with a 95% CI of 0.005–0.53% and 0.17% with a 95% CI of 0.008–0.92%, respectively (Table 2). We also conducted an experiment using 5 groups of 200 snails (1000 snails in total) simultaneously at the P2 breeding sites. The same positive rate of 0.10% (0.005–0.56%) was obtained in an experiment with 20 groups of 50 snails. In 2018, we analyzed 20 groups (1000 for each site) at 3 breeding sites (P2, P4, and P5). No positive group was found at any of the breeding sites (0/3000). 

### 3.4. Heat Map of Infected Residents and Infected Snail Sampling Location and the Relationship between Infection and Lifestyle Habits

We plotted the location of schistosomiasis patient residences on a map, together with those of infected snail habitats from the 2016 survey. The map showed that the patients’ residences were located near the infected snail habitats (Figure 3A). In the Khon Neua and Hang Khong villages, no *S. mekongi*-infected snails were found, although infected patients lived there in 2016. A heat map was created based on the infection rates of both host snails and humans (Figure 3B). It suggested that the distribution of infected host snails had a strong influence on the risk of *S. mekongi* infection in humans. Finally, an interview survey was conducted to examine the characteristic lifestyles of residents determined as infected via LAMP assays. We found that the absence of a latrine in their house and the use of well water were correlated with *S. mekongi* infection (Table 3). This suggests that people living near infected snails are at risk of infection through contact with contaminated water.

## 4. Discussion

Lao PDR is classified as a low-schistosomiasis-transmission country, meaning that less than 1% of high-intensity infections are recorded at sentinel sites, which has been the case since 2017. The next goal is to eliminate schistosomiasis transmission by 2030 [8]. In areas with low transmission, the occurrence of an outbreak following MDA discontinuation represents a challenge for eliminating schistosomiasis. It is also necessary to monitor and control the re-introduction of schistosomiasis, even after its elimination [25,26]. However, in Laos, the current monitoring system is limited to detecting mild infection, as only the Kato–Katz stool examination method is employed for the detection of *Schistosoma mekongi* infection throughout the country. Therefore, highly sensitive diagnostic methods, such as LAMP and PCR assays, are urgently needed. The LAMP assay has been widely employed for the detection of various parasitic diseases, becoming a standard molecular diagnostic method due to its greater sensitivity, convenience, and cost efficiency when compared to conventional PCR [2,27,28,29,30]. Some studies have employed LAMP for the control of schistosomiasis [13,19,31].

This is the first study in Lao PDR to employ LAMP for the detection of *S. mekongi* DNA in stool samples from residents and from snail samples of the intermediate host *Neotricula aperta*. The LAMP method detected infected fecal samples with higher sensitivity than the Kato–Katz method. This is in agreement with reports on other parasitic diseases, which also confirmed the superior sensitivity achieved via LAMP [27,28,29,30]. DNA extraction using the heat alkaline method is a simple, time-efficient, and low-cost method. However, as this method extracts a large amount of DNA from feces, it was also confirmed that diluting the sample with water yields better results. We are confident that the DNA extraction and LAMP methods employed herein are worthy of becoming the standard for fecal examination. 

According to previous studies, the *S. mekongi* infection rate among *N. aperta* was reported to be very low (<0.3%) [2,32,33]. Therefore, large-scale sampling of host snails was performed annually in the endemic area from 2016 to 2018. In 2016, a pool of 200 snails was subjected to LAMP assays. *S. mekongi*-infected snails were present at the three breeding sites (P1, P2, and P3), whereas infected snails were detected in all groups at the other two breeding sites (P4 and P5). Thus, it was difficult to precisely estimate the infection rate at the two breeding sites (P4 and P5). Therefore, in 2017, DNA extraction was performed using a pool of 50, instead of 200, snails. The results of the analysis using 20 groups of 50 snails were in agreement with observations made by checking each snail individually. The infection rate analyzed in 20 groups of 50 snails was consistent with that determined via the analysis of 5 groups of 200 snails. These results suggest that in areas where the infection rate is low, an experiment using 50 snails may be a good fit for DNA detection-based monitoring. Indeed, in areas where the infection rate is about 0.1% and there is a large population of snails, it may be more appropriate to consider pooling 200 snails.

Heat mapping of *S. mekongi* infection areas based on the 2016 LAMP results revealed that most schistosomiasis patients lived close to the breeding sites of infected host snails in the Khon Tai village. These results suggest that the distribution of infected host snails has a strong influence on the risk of *S. mekongi* infection in humans. In fact, a number of previous studies reported a close association between schistosomiasis infection and water contact [34,35,36]. However, *S. mekongi*-infected snails were not detected in the Khon Neua village, although four schistosomiasis patients (4/96) and many *N. aperta* (*n* = 1150) were recorded in the village. In contrast, in the Hang Khon village, no *N. aperta* were found along the riverside in 2016 and 2017, although one schistosomiasis patient (1/87) was reported in 2016. One possible explanation for these results is that people move or travel more frequently and at greater distances than their host snails. Thus, *S. mekongi* human infections may occur at places that are not near the residence of the affected person, but also in other locations, e.g., during fishing. 

Furthermore, as a result of our questionnaire survey, some of the participants’ lifestyles were found to be associated with *S. mekongi* infection. For example, no latrines and the use of a well at home were significantly correlated with the risk of *S. mekongi* infection (Table 3). However, it should be noted that our survey addressed the use of latrines with a “yes” or “no” question. Therefore, there were no data on latrine usage among the participants. It also remains unknown why the use of wells at home is correlated with *S. mekongi* infection risk. 

However, several studies have suggested that the availability of water, sanitation, and hygiene (WASH), in addition to MDA, is likely to contribute to the prevention of schistosomiasis transmission [26,35,37,38,39,40]. In the case of *S. haematobium*, which is distributed in Africa, the use of water from an open source for household affairs, such as cleaning and bathing/swimming, was associated with a higher prevalence of *S. haematobium* infection, while the infection rate was significantly lower among those who preferred using latrines [41]. Furthermore, frequent contact with unprotected water sources, not using latrines, and the lack of information on schistosomiasis were suggested to predispose school children to *S. haematobium* infection [42]. Our data support the conclusion that hygiene, especially regarding latrine use, is important in the prevention of *S. mekongi* infection, as for other schistosomiasis infections. 

This is the first study to develop and evaluate a LAMP assay for detecting *S. mekongi* DNA in human stool and host snail samples in elimination settings, that is, low-transmission areas of schistosomiasis mekongi. Our LAMP assay was proven to be highly sensitive compared with Kato–Katz stool examination, while also offering greater ease of handling and cost efficiency when compared to conventional PCR. Taken together, the LAMP assay represents an effective molecular tool for monitoring schistosomiasis mekongi prevalence through the analysis of human stool and host snail samples to then establish infection risk maps.

## Figures and Tables

**Figure 1 pathogens-11-01413-f001:**
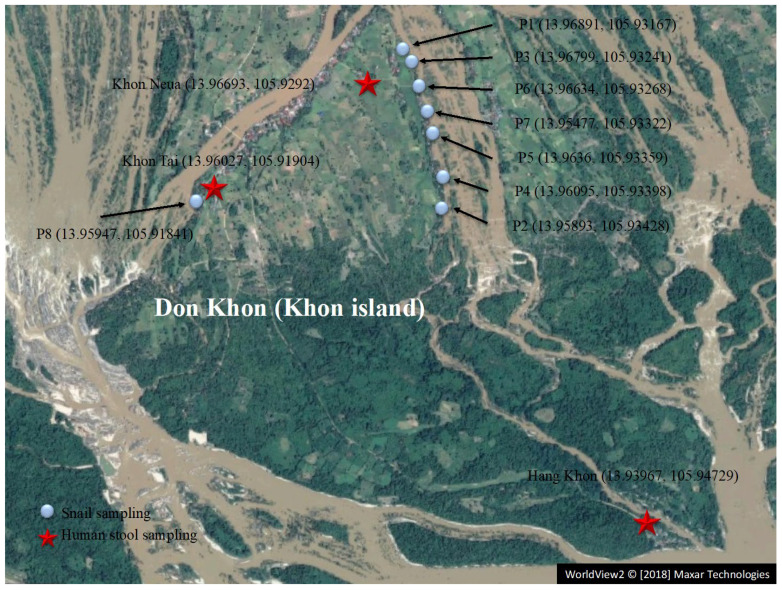
Study areas of Don Khon (Khon Island), Champasak Province, Laos. A map of Khon Island is also presented. Red stars show the locations where stool samples were collected from the study participants in 2016. Light blue circles show sampling sites of the host snail *N. aperta* from 2016 to 2018. Latitude and longitude are shown in parentheses. Background satellite image: WorldView2 © [2018] Maxar Technologies.

**Figure 2 pathogens-11-01413-f002:**
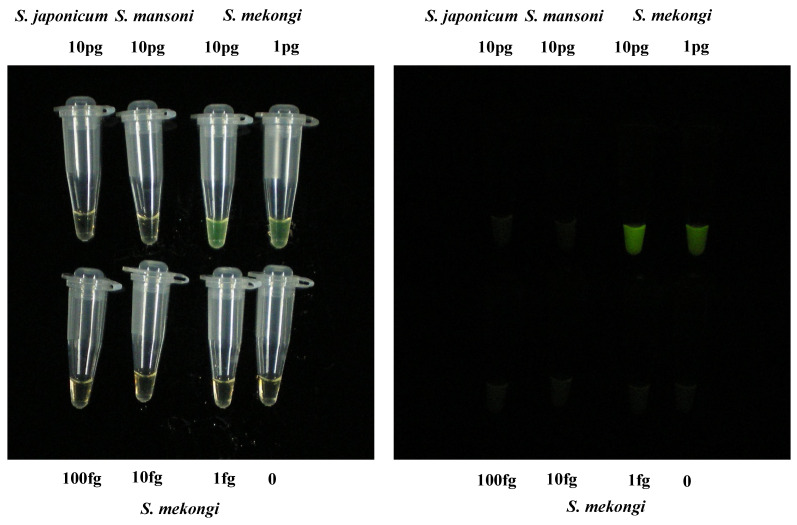
Sensitivity and specificity of LAMP assay that targets the ITS1 gene of *S. mekongi*. The LAMP assay was able to detect 1 pg of *S. mekongi* DNA, but neither *S. mansoni* nor *S. japonicum* DNA was detected. The left panel shows the LAMP results under daylight. The right panel shows LAMP results obtained under UV light. Fluorescence (light green) was confirmed in the reaction tubes containing *S. mekongi* DNA (10 pg and 1 pg).

**Figure 3 pathogens-11-01413-f003:**
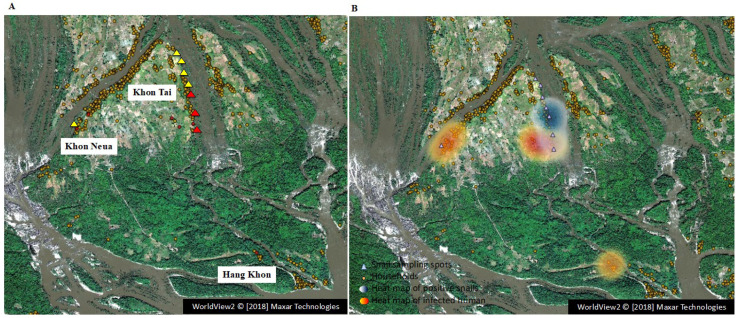
Mapping of LAMP-positive residents (schistosomiasis mekongi patients) and host snails, yielding a heat map of *S. mekongi* infection risk. (**A**) All houses in Khon Island were plotted as circles on the map. The snail sampling sites were also plotted as triangles (in 2016–2018). Locations of LAMP-positive residents and host snails are shown in red on the map (LAMP data in 2016). (**B**) A heat map was made using the kernel density estimation based on the data of LAMP-positive residents (red) and host snails (blue). Background satellite image: WorldView2 © [2018] Maxar Technologies.

**Table 1 pathogens-11-01413-t001:** Comparison of LAMP method and Kato–Katz method using stool samples collected from residents of Khon Island in 2016.

	LAMP Method		Kato–Katz Method									
	*Schitosoma mekongi*		*S. mekongi*		*Opisthorchis viverrini*		Hookworms		*Trichuris trichiura*		*Taenia* sp	
Village	Positive	Negative	Positive	Negative	Positive	Negative	Positive	Negative	Positive	Negative	Positive	Negative
Khon Neua	4 (4.2%)	92	0	96	48 (50.0%)	48	22 (22.9%)	74	1 (1.0%)	95	2 (2.1%)	94
Khon Tai	3 (3.4%)	86	1 (1.1%)	88	59 (66.3%)	30	20 (22.5%)	69	0	89	5 (5.6%)	84
Hang Khon	1 (1.1%)	86	0	87	45 (51.7%)	42	20 (23.0%)	67	3 (3.4%)	84	0	87
Total	8 (2.9%)	264	1 (0.4%)	271	152 (55.9%)	120	62 (22.8%)	210	4 (1.5%)	268	7 (2.6%)	265

**Table 2 pathogens-11-01413-t002:** Results of the snail survey on Khon Island using a large-scale LAMP method from 2016 to 2018.

Year	2016				2017				2018		
Spot	Total Number of Examined Snails	Number of Positive Groups/Number of Total Groups		Infection Rate (95%CI) (%)	Total Number of Examined Snails	Number of Positive Groups/Number of Total Groups		Infection Rate (95%CI) (%)	Total Number of Examined Snails	Number of Positive Groups /Number of Total Groups	Infection Rate (95%CI) (%)
		Group Size of Snails Crushed Together				Group size of Snails Crushed Together				Group Size of Snails Crushed Together	
		176 or 86	200			50	200			50	
P1	200	-	0/1	0 (0~1.49)	-	-	-	-	-	-	-
P2	1400	-	3/7	0.28 (0.07~0.74)	1000	1/20	-	0.10 (0.005~0.56) ^a^	1000	0/20	0 (0~0.36)
P2	-	-	-	-	1000	-	1/5	0.11 (0.005~0.53) ^b^	-	-	-
P3	176	0/1	-	0 (0~1.69)	-	-	-	-	-	-	-
P4	400	-	2/2	not calculated—(0.13~) ^c^	600	1/12	-	0.17 (0.008~0.92)	1000	0/20	0 (0~0.36)
P5	86	1/1	-	not calculated—(0.06~) ^c^	700	0/14	-	0 (0~0.54)	1000	0/20	0 (0~0.36)
P6	600	-	0/3	0 (0~0.50)	400	0/8	-	0 (0~0.90)	-	-	-
P7	-	-	-	-	1050	0/21	-	0 (0~0.35)	-	-	-
P8	-	-	-	-	1150	0/23	-	0 (0~0.31)	-	-	-
Total	2862				5900				3000		
				^c^: 95%CI lower bound				^a^: estimated value by using a group size of 50 ^b^: estimated value by using a group size of 200			

The light gray background shows the experiments of 50 crushed snails, and the dark gray background shows those of 200 crushed snails.

**Table 3 pathogens-11-01413-t003:** Socio-economic status and lifestyle of residents with respect to *Schistosoma mekongi* infection diagnosed by LAMP method.

Variables	Total	Negative	Positive	*p*-Value
	(*n* = 275)	(*n* = 267)	(*n* = 8)	
**Gender ^1^**	□	□	□	□
Female	164	160	4	0.414
Male	111	107	4	□
**Age ^1^**	□	□	□	□
Child (≤18)	53	53	1	0.508
Adult	220	212	7	□
**Educational status ^1^**	□	□	□	□
No grade completed	42	40	2	0.849
Primary school	135	131	4	□
Secondary school	96	94	2	□
Higher	2	2	0	□
**Occupation ^1,2^**	□	□	□	□
Agriculture	179	173	6	0.667
Housewife	20	19	1	□
Other	18	18	0	□
**Village ^1^**	□	□	□	□
Khon Neua	96	92	4	0.442
Khon Tai	89	86	3	□
Hang Khon	90	89	1	□
**Number of people in the household (mean SD) ^3^**	6.1 (2.3)	6.3 (2.2)	6.0 (2.7)	0.753
**Family use of Mekong River ^1^**	□	□	□	□
No	32	29	3	0.054
Yes	243	238	5	□
**Well ^1^**	□	□	□	□
No	252	247	5	0.022 *
Yes	23	20	3	□
**Rainwater storage ^1^**	□	□	□	□
No	268	260	8	0.811
Yes	7	7	0	□
**Purchase of bottled water ^1^**	□	□	□	□
No	213	205	8	0.126
Yes	62	62	0	□
**Latrine^1^**	□	□	□	□
No	72	67	5	0.031 *
Yes	203	200	3	□
**Enter Mekong River ^1^**	□	□	□	□
No	48	45	3	0.147
Yes	227	222	5	□

^1^ Fisher’s exact test between the results of parasitic infection by LAMP. ^2^ Only participants over 20 years old (*n* = 212). ^3^ ANOVA between the results of parasitic infection by LAMP. * S.D.

## Data Availability

Not applicable.

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
