# Peer review of "Detection of Schistosoma mekongi DNA in Human Stool and Intermediate Host Snail Neotricula aperta via Loop-Mediated Isothermal Amplification Assay in Lao PDR"

_pathogens, 2022, doi:10.3390/pathogens11121413_

Round 1

Reviewer 1 Report

Only minor comments

Line 212; reference or refer to Table 1

Line311; delete identified

Line 312; agreement

Line 322 S. mekongi

Lines 355, 357, 361; S. haematobium

Author Response

Response to Reviewer 1

Thank you for your reviewing. I am glad you agree with the content of the paper.

Line 212; reference or refer to Table 1

Line311; delete identified

Line 312; agreement

Line 322 S. mekongi

Lines 355, 357, 361; S. haematobium

The minor point has been corrected; I wasn't sure what you pointed out about L212, is it the location of Table 1 ?

Reviewer 2 Report

General comments:

This paper describes a LAMP assay for Schistosoma mekongi using a simple DNA extraction method. In particular, the paper evaluated the utility of the LAMP assay for detecting S. mekongi DNA in human stool and snail samples in endemic areas in Laos. The results showed that 0.4% (1/272) of the stool samples were positive for S. mekongi eggs, as opposed to 2.9% (8/272) positive for S. mekongi DNA based on LAMP assays. The paper indicate that the LAMP assay is an effective method for monitoring pathogen prevalence and creating risk maps for schistosomiasis.

  My reservations are as follows:
(1) It is valuable to develop accurate and sensitive methods of diagnosis capable of detecting very light infections in a significant decrease in the prevalence and intensity of the Schistosomiasis.    

Although the work showed that the LAMP assay seemed superior sensitivity method for the detection of Kato-Katz technique in detecting low-intensity S. mekongi infections, the positive rates were only 2.9% (8/272). I felt that the positive cases were too small and the results were of too preliminary a nature for publication.  

(2) It should evaluated the utility of the LAMP assay for detecting S. mekongi DNA in human serum samples. Because the LAMP assay for detecting the DNA of serum samples may be much more superior sensitivity of LAMP in detecting the DNA of stool samples. (reference paper: J. Xu et al. Sensitive and rapid detection of Schistosoma japonicum DNA by loop-mediated isothermal amplification (LAMP). International Journal for Parasitology 40 (2010) 327–331)

The additional data to demonstrate that superior sensitivity of LAMP in detecting very light infections of S. mekongi infections would strengthen the manuscript considerably.

Author Response

Response to Reviewer 2

Thank you for your reviewing. We appreciate your interest in this paper and your precise comments.

(1) It is valuable to develop accurate and sensitive methods of diagnosis capable of detecting very light infections in a significant decrease in the prevalence and intensity of the Schistosomiasis. Although the work showed that the LAMP assay seemed superior sensitivity method for the detection of Kato-Katz technique in detecting low-intensity S. mekongi infections, the positive rates were only 2.9% (8/272). I felt that the positive cases were too small and the results were of too preliminary a nature for publication.  

Currently, the infection rate in Laos is decreasing due to measures taken by WHO, and it is becoming more difficult to find places with positive patients. We report that our method is useful for monitoring infection in such a low infection rate situation with a highly sensitive LAMP method, which we believe is necessary to prepare for emergence.

(2) It should evaluated the utility of the LAMP assay for detecting S. mekongi DNA in human serum samples. Because the LAMP assay for detecting the DNA of serum samples may be much more superior sensitivity of LAMP in detecting the DNA of stool samples. (reference paper: J. Xu et al. Sensitive and rapid detection of Schistosoma japonicum DNA by loop-mediated isothermal amplification (LAMP). International Journal for Parasitology 40 (2010) 327–331)

You know that the cited study is a paper targeting the retrotransposon of Schistosoma japonicum. The same sequences targeted in S. mekongi have not been identified. In addition, a recent paper reported that DNA detection in serum of S. mekongi is inadequate and less specific (added in ref. 11). Furthermore, since non-invasive specimen collection is highly appreciated in local diagnosis, DNA extraction from feces was employed. Since this result is more sensitive than the Kato-Katz method, we believe it can be used adequately for surveillance.

Reviewer 3 Report

Schistosmiasis mekongi is locally distributed in Laos and Cambodia. After many rounds of MDA, the prevalence of schistosomiasis decreased to a very low level, thus needs more sensitive tools for identifying cases or risk environments. This article explored the possibility of LAMP technique for detecting pathogens of S. mekongi in residents and snails. I have several comments on this article for considereation

1.        Background: Is not sufficient to support the research objectives. Is there any other molecular method for S. mekongi? How about the performance

2.        Line 112, “per previous surveys” should be per previous survey

3.        The research is not planned well and lack of systematic design, such as the years of survey, the location selected and the control method set

4.        Please notice that “S. mekongi” should be italic, please modify through the manuscript.

5.        In the method of LAMP, please concrete the reaction system and which kind of chromogenic agent used

6.        Please reorganize the tables especially table 2 which confused the reader easily

7.        In the part of result, do not mix the methods and discussion in this part, just describe the research findings

8.        Why the snails pooled in different ratio? As no previous evidence support these ratios based on laboratory assessment, I am afraid that the results are unpersuasive and the calculation of infection rate of snails is also unscientific.

Author Response

Response to Reviewer 3

Thank you for your reviewing. We appreciate your interest in this paper and your precise comments.

  1. Background: Is not sufficient to support the research objectives. Is there any other molecular method for S. mekongi? How about the performance

Very few papers on molecular diagnostics have been published in Schistosoma mekongi, unlike other Schistosoma spp. Previous papers performed using real-time PCR in the laboratory and recently published papers have been added to the introduction. However, we would like to emphasize that our study is the first paper on the LAMP method using field samples.

  1. Line 112, “per previous surveys” should be per previous survey

Corrected as noted.

  1. The research is not planned well and lack of systematic design, such as the years of survey, the location selected and the control method set.

Information on the island chosen as the study area and the reason for selection was added to the Method. In addition, the survey was conducted over a three-year period as indicated in the Results. The project has now been completed and the results are the data obtained during the last three years.

  1. Please notice that “S. mekongi” should be italic, please modify through the manuscript.

Corrected as noted.

  1. In the method of LAMP, please concrete the reaction system and which kind of chromogenic agent used

Fluorescent Detection Reagent (Eiken Science, Tokyo, Japan)was used as the fluorescent reagent for the LAMP assay, and a kit from the same company was used for the reaction. This was added to the method.

  1. Please reorganize the tables especially table 2 which confused the reader easily

The results in Table 2 present all the information in this study. We have color-coded them because they are difficult to understand. The information in the P2 area is particularly important, and we would like to draw your attention to this area. Also, this is a study of how many snails are taken and in what groups to treat to measure infection rates, and these are the results that will form the basis for future methods of detecting DNA from pooled snails.

  1. In the part of result, do not mix the methods and discussion in this part, just describe the research findings

Deleted last line of results. I think it basically only mentions what is based on the results.

  1. Why the snails pooled in different ratio? As no previous evidence support these ratios based on laboratory assessment, I am afraid that the results are unpersuasive and the calculation of infection rate of snails is also unscientific.

At the beginning of our research, we did not know how many snails are infected with S. mekongi. For this reason, we initially conducted an experimental system with one group of 200 snails. However, this did not provide an accurate measure of the infection rate. So we found that we could measure the infection rate by making 20 groups of 50 snails in one group. Please confirm the detailed calculation method in Results. This method of pooling snails and extracting DNA is also used in our LAMP method for S. japonicum, and the calculation using 50 snails has already been published (Ref. 13). Therefore, there is a scientific basis for this method.

Round 2

Reviewer 2 Report

The revision of Manuscript and response are well.